# Feeding activity of *Eisenia andrei* and *Enchytraeus albidus* under different soil moisture regimes assessed by the bait-lamina test

Gilda Dell'Ambrogio[1¤a]*, Sophie Campiche[1¤b], Janine W. Y. Wong[1], Mathieu Renaud[1], Christina Lüthi[1], Inge Werner[2], Benoit J. D. Ferrari[1,2]

1 Swiss Centre for Applied Ecotoxicology, Lausanne, Vaud, Switzerland, 2 Swiss Centre for Applied Ecotoxicology, Dübendorf, Zürich, Switzerland

¤a Current address: Environmental Analytics, Methods Development and Analytics, Agroscope Reckenholz, Zürich, Switzerland
¤b Current address: EnviBioSoil, Gollion, Vaud, Switzerland
* gilda.dellambrogio@agroscope.admin.ch

## Abstract

The bait-lamina test is an easy and efficient method to quantify the feeding activity of soil invertebrates. However, under natural conditions, feeding activity is influenced by environmental factors such as soil moisture content, whose effects on test outcomes remain insufficiently quantified, complicating interpretation. In this study, we optimized the bait-lamina test under laboratory conditions, to assess the influence of soil moisture content on feeding activity of the earthworm *Eisenia andrei* and the enchytraeid *Enchytraeus albidus*, using LUFA 2.2 standard soil (LUFA Speyer, Speyer, Germany) as a substrate. Feeding activity increased linearly with increasing soil moisture until an optimal moisture content, which was 52% of the maximum water holding capacity (WHC) for earthworms, and between 49 and 68% WHC for enchytraeids. Above these optima, feeding activity was reduced and was less dependent on soil moisture. The increase in feeding activity up to the peak was described by different slopes for the two species. Earthworms consumed the bait faster (on a per unit weight basis) than enchytraeids. Among the two species, the relationship established for *E. albidus* was the more similar to the response obtained in field conditions. Within the range of soil moistures considered by the model, our results demonstrate that feeding activity is positively correlated with soil moisture for two important soil invertebrates, although this increase, the optimal soil moisture content, and the speed of bait consumption are species-dependent. The model produced provides a first quantitative framework describing these relationships and can serve as a basis for future studies testing its applicability across different soil types. Such research would represent a first step for normalizing results from bait lamina tests under field conditions.

**Data availability statement:** All relevant data are within the paper and its Supporting information files.

**Funding:** The author(s) received no specific funding for this work.

**Competing interests:** The authors have declared that no competing interests exist.

## Introduction

Soil organisms play an essential role in the breakdown of organic matter, which is a fundamental step in the cycling and regulation of nutrients in soil. Among the available tools to assess soil ecological and biological parameters, the standardized bait-lamina method [1] is a simple functional test allowing the *in-situ* measurement of the feeding activity of soil organisms. Feeding activity, an indicator of biological activity, is evaluated at the community level by recording the consumption of organic bait provided in perforated plastic strips inserted vertically into the soil [2,3]. Earthworms and enchytraeids are considered to be the main groups of soil organisms feeding on the bait while the role of micro-arthropods and microorganisms appears to be less important [4–6].

The bait-lamina test can be applied to assess the effect of chemicals on soil organisms, by comparing feeding activity rates between sites of interest and reference sites [7,8], or for the long-term monitoring of soil biological quality [9,10]. The test is nowadays one of the recommended tools for the ecological risk assessment of contaminated sites [11,12]. First developed as a field method, the bait-lamina test has also been described as a promising screening tool for assessing effects of chemicals in the laboratory [7,13–16].

Under field conditions, abiotic factors such as soil moisture and temperature can influence soil faunal activity and therefore complicate the interpretation of the results of the bait-lamina test [3,17]. Nevertheless, only a few studies have explored how these factors influence feeding activity. Gongalsky et al. [4] found that feeding activity of enchytraeids increased with increasing soil temperature, while the influence of soil moisture was less evident. Other studies observed that soil moisture was one of the main abiotic factors driving biological activity, suggesting that the first positively influenced the latter [5,18,19]. However, the data generated in these experiments did not allow a detailed characterization of the statistical relationship between the two factors.

In a field study performed by Campiche et al. [20], the bait-lamina assay was used to compare feeding activity across four plots combining two crop types, two manure types, and herbicide treatment (with/without). The method was applied at two time points: before and after seedling. When pooling all the replicates without herbicide treatment, the authors observed an increase in feeding activity with increasing soil moisture content. However, the produced model was based on a small number of data points (8 treatments with 5 replicates each). Additional studies are needed, to establish a more robust model, which could be applied to normalize data between tests. Such a model would represent an important tool for comparing bait-lamina test results obtained under field conditions, where soil moisture may vary considerably between sites, and for improving the interpretation of field data.

The present study applied the bait-lamina test in the laboratory using two model organisms for ecotoxicity testing, the earthworm *Eisenia andrei* and the enchytraeid *Enchytraeus albidus*. The aim is to investigate the feeding behavior of these two model species, and especially to characterize the relationship between soil moisture content and their feeding activity.

## Materials and methods

### Preparation of the bait-lamina strips

The bait-lamina PVC strips were 160 mm long, 6 mm wide and 1 mm thick, and perforated with 16 bi-conical apertures 5 mm apart. They were purchased from Terra Protecta (GmbH, Berlin, Germany). The bait substance consisted of a powdery mixture of cellulose, wheat bran flakes and activated charcoal at a ratio of 70:25:5 (w:w:w), which was mixed with Milli-Q water (ratio water to bait of approx. 1.4:1, v:w) to form a paste [1]. Filled bait-lamina strips were air-dried for at least 24 h and checked for complete filling of bait-lamina perforations prior to each test.

### Test organisms

Based on preliminary research, two species routinely tested in ecotoxicological bioassays were chosen for the bait-lamina tests, i.e., the earthworm (*Eisenia andrei*), and the enchytraeid (*Enchytraeus albidus*).

*Eisenia andrei* Bouché 1972 (Annelida: Oligochaeta) were originally obtained from the farm Lombritonus (Ollon, Switzerland; http://www.lombritonus.ch) and maintained in laboratory cultures. The breeding substrate was composed of a moist mixture of fresh horse manure, composted manure and peat moss in a proportion of 1:1:1 (v:v:v) and fed approximately once a week with finely ground rolled oats previously heated at 105 °C for 48 h.

*Enchytraeus albidus* Henle 1847 (Annelida: Oligochaeta) were originally obtained from the French National Institute of Agricultural Research (INRA/AgroParisTech, UMR ECOSYS, Versailles, France) and maintained in laboratory cultures. The breeding substrate consisted of a moist mixture of potting soil and LUFA 2.2 at a ratio of 3:2 (v:v). Organisms were fed twice a week with rolled oat flakes and cat food pellets, both previously heated at 105 °C for 48 h and finely ground.

Both cultures were maintained at 20±2 °C in the dark for several generations before their use in experiments.

### Soil used in experiments

The soil used in all experiments was the standardized LUFA 2.2 soil (LUFA Speyer, Speyer, Germany) a natural sandy loam soil, according to USDA particle size distribution (%), with properties provided by the producer: pH 5.4, total organic carbon 1.59%, clay 7.7%, silt 16.2%, sand 76.1% and 45.8±1.9% maximum water holding capacity (WHC). For the experiments, the soil moisture content of the LUFA 2.2 soil was measured in triplicates by oven drying at 105 °C. Gravimetric moisture content was calculated as the water mass per unit dry soil mass and converted to a percentage of maximum water holding capacity (relative moisture content, % WHC) to enable future comparison with other soils and to facilitate the link with the optimal moisture conditions for ecotoxicity testing, usually expressed in the same terms.

In previous pilot studies it was observed that after five days, the bait was partially consumed in LUFA 2.2 soil without the test organisms. To prevent any infestation (community development) by unwanted soil organisms, the soil used for the test lasting 12 days with *E. albidus* was defaunated by freezing at −21 °C for six days prior to test start. For tests lasting only two days (*E. andrei*), the soil was not defaunated because no infestation of the medium was observed in this short lapse of time. To further verify the absence of any additional contribution to bait consumption other than the tested organisms, negative control replicates were run for both species without organisms, as detailed below.

### Experimental design

For each experiment, polystyrene plastic containers (171 x 123 x 60 mm) were filled with LUFA 2.2 soil pre-moistened at different range of humidity. Test organisms and bait lamina were then added to the soil.

For earthworms, the experiment was conducted in two series. The first included nominal relative moisture contents of 20, 40, 60, 80, and 100% of the WHC. The second added additional levels around the reference moisture content (30, 45, 60, 75, 90, and 105% WHC) to refine the gradient. Five replicates were made for each moisture treatment. The nominal 60% WHC represents standard testing conditions for earthworms in ecotoxicology [21–23] and was thus considered as

"reference" to allow comparison with the other moisture treatments, as well as between the two test series. Three negative control replicates were run at 60% and at 100% WHC without earthworms, to further verify if the bait was consumed in the absence of the test organisms.

For the enchytraeid experiments, feeding activity was expected to respond within a narrower moisture range, as suggested by preliminary observations. Therefore, a single test was conducted with five nominal moisture contents (40, 50, 60, 70, and 80% WHC). Given the larger number of individuals required per replicate compared to earthworms (see below), replication was adjusted to four per treatment. The reference moisture content was set at 50% WHC, representing standard testing conditions for enchytraeids in ecotoxicology [24,25]. Two negative control replicates containing no enchytraeids were included for each moisture treatment.

The gravimetric soil moisture content was measured for each moisture treatment, at the beginning and end of the test, based on a composite sample from all replicates.

The number of organisms per replicates was chosen based on the geometric mean of several population densities, collected from the literature for arable soils in Switzerland and neighboring countries (see S1 and S2 Tables in S1 File for details). Moreover, to allow a comparison between the feeding rates, a similar biomass, i.e., wet weight of the total number of individuals per replicate, was used for the two species.

For the earthworm experiments, five organisms were placed into 900 g of moist LUFA 2.2 soil for each replicate, corresponding to a field density of 225 individuals m$^{-2}$. Only adults aged between 2 and 8 months and with an individual wet weight between 300–600 mg were used for the tests. Based on the range of individual earthworm weight used, the estimated total average biomass of five individuals per replicate was 2.25 ± 0.75 g. The selected worms were acclimated for 48 hours in a separated container filled with LUFA 2.2 soil moistened at 60% WHC, prior to the tests.

For the enchytraeid experiments, 230 organisms of similar size and at least 1 cm in length were introduced into 600 g of moist LUFA 2.2 soil per replicate, corresponding to a field density of 11'000 individuals m$^{-2}$. The mean wet weight of a single organism was estimated at 0.010 ± 0.003 g, based on repeated measurements of samples consisting of a defined number of individuals. Based on this, the estimated total average biomass of 230 individuals per replicate was 2.30 ± 0.69 g, which is comparable to the earthworm biomass used in the test. *E. albidus* were extracted from the culturing substrate by a wet extraction adapted from the annex A of the ISO guideline 16387 [25]. Briefly, approximately 500 ml of culture soil was carefully crumbled and placed into a metal sieve (180 mm diameter, mesh size 1–2 mm). The bottom of the sieve was previously covered with a perforated aluminum foil to avoid soil particles passing through the sieve. The sieve was hung in a plastic bowl, which was filled with tap water until it covered the soil completely. Because of gravity and of their constant motion, the enchytraeids passed through the perforated aluminum foil and the sieve and fell into the water. Water was kept cold with an ice pack and constant aeration was provided with a pump, since a lack of oxygen could cause mortality to the worms [26]. The extraction duration ranged from 6 to 13 h to minimize the time the worms spent in the water. Longer periods were found to cause excessive stress in the animals (i.e., lower mobility, higher mortality). At the end of the extraction procedure, the greater part of the water present in the bowl was slowly decanted; the worms were rinsed with tap water and immediately transferred to petri dishes for counting. Before starting the experiment, enchytraeids were acclimated in the test vessels for 48 h, after which the bait-lamina strips were added.

For both earthworms and enchytraeids experiments, five bait-lamina strips were inserted horizontally into each test container, just beneath the surface of the soil and about 2 cm apart (Fig 1).

The test containers were subsequently covered with a 500 μm mesh cotton fabric held in place with a perforated lid to prevent escape of the organisms. Test duration was determined in pilot studies and defined as the time required to reach a mean feeding activity of more than 30% in the reference replicates [1]. This took 2 days for earthworms and 12 days for enchytraeids. All tests were conducted at 20 ± 2 °C with a 16:8 h light-dark photoperiod, according to standard testing and culture conditions for ecotoxicity testing [21–24].

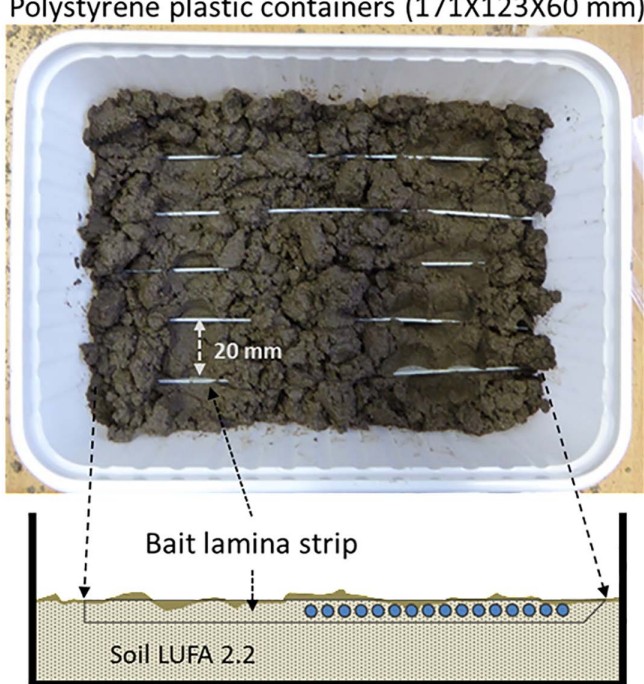

**Fig 1. Test system for exposing organisms to five bait-lamina strips.**

For the enchytraeid test, which had a longer duration compared to earthworms, soil water content was checked periodically by weighing the test containers and Milli-Q water was added when a weight loss of more than 2% was observed [24]. The moisture content of the earthworm tests was not monitored, as the tests lasted only 2 days.

At test termination, bait-lamina strips were carefully removed from the soil and gently rinsed with tap water [3,7]. Feeding activity was assessed according to ISO 18311 [1]. Food consumption on the bait-lamina strips was categorized by attributing a score of 2 to a completely empty hole, 1 to a partly empty hole, and 0 to a hole with bait intact. Feeding activity was expressed as percentage of pierced holes per strip (percentage of bait consumed).

Surviving organisms were retrieved from the test soil and their numbers recorded. Living earthworms were extracted by hand, while survival rate of enchytraeids was assessed by wet sieving extraction [27]. Briefly, 3 L of tap water was added to the soil of each replicate, and enchytraeids were fixed with an aqueous solution of 4% (v:v) formaldehyde. Subsequently, the wet soil was washed 7–8 times through a 355 μm mesh sieve and the extracted enchytraeids were stained with a few mg of Bengal rose and transferred to a petri dish for counting.

## Statistical analyses

To allow a better comparison between the different experiments, results of feeding activity for both species were converted into daily feeding activity by dividing the overall feeding activity by the test duration in days. All data were checked for normality and homogeneity of variance using the Shapiro-Wilk and the Levene test, respectively. An analysis of variance followed by a Dunnet's post hoc test with Holm's correction for multiple comparisons was used to compare the different treatments with the reference response, and an unpaired two samples t-test was used for a comparison between two treatments only. To investigate the correlation between feeding activity and soil moisture content, Spearman's test was used. Based on the results obtained, further analysis focused only on data showing a strong and clear correlation, i.e.,

from lowest to the peak moisture content, omitting results from treatments above the optimum moisture content of each species. Correlation between daily feeding activity and moisture within this range was analyzed by means of simple and multiple linear regressions. First, two simple linear regressions were fitted, one for each species. Secondly, differences between the two experiments were investigated by means of a multiple linear regression with the fixed effects moisture and species, plus the interaction between moisture and species. Multiple linear regressions were then also used to compare the linear regressions produced in this study with the ones obtained a previous field study from Campiche et al. [20]. From the field study, the maximum WHC of the field soil was not available and moisture contents were given in gravimetric units (percentage of the soil dry weight). Therefore, for this last step, moisture contents were always expressed in gravimetric units for comparison. All statistical tests and linear regression models were performed with the R software within the R studio environment.

## Results

### Test performance

Overall mortality rate for both species was low (< 10%), except for the earthworm treatment at 25% WHC, where mortality was 12% (see SI, Section 2). Outside the optimal moisture commonly used for testing (40 – 60% WHC) effects and stress cannot be excluded and could explain the slightly larger mortality in such dry treatment. At the end of the test, average feeding activity in reference moisture treatments met the validity criteria (≥ 30%) required for the controls in the standard testing guideline [1]. Negative control replicates without test organisms showed no bait consumption at all moisture treatments, except for the nominal 80% WHC (enchytraeid test) and 100% WHC (earthworm test), where the average daily feeding activity was 1.04 ± 1.30 and 0.08 ± 0.04 (% of consumed bait per day ± SD), respectively (all data in S3 Table in S1 File).

Measured soil moisture content was generally slightly lower than the nominal values and declined further by the end of the test, with these differences being more pronounced in the earthworm experiments (see S3 Table and S4 and S5 Figs in S1 File). Because of these differences, the average of the measured values between the start and the end of the test were used for the data analysis, instead of the nominal values. For the two tests on earthworms, the results of feeding activity at the two reference moisture contents (nominal 60% WHC) were not significantly different (unpaired two samples t-test > 0.05). Therefore, the two tests were pooled and the two reference moisture treatments (measured 53% and 51% WHC) were merged as a unique treatment, corresponding to the average of the two (measured 52% WHC). Similarly, the respective values of feeding activity were also pooled.

### Soil moisture effect on feeding activity

The results of feeding activity for the different soil moisture contents are provided in S3 Table in S1 File and depicted in Fig 2. For the earthworms pooled test (Fig 2A), the average daily feeding activity was the highest (21.82 ± 10.41% of consumed bait per day ± SD) at the reference relative moisture content (52% WHC) and were significantly lower at all other moisture treatments.

For enchytraeids (Fig 2B), the highest average daily feeding activity (7.62 ± 0.39% of consumed bait per day ± SD) occurred at a relative moisture content higher than the reference (68% rather than 49% WHC), although the difference in result between the two moisture conditions was not statistically significant. Average daily feeding activity at 40% WHC and 78% WHC was significantly lower compared to the reference moisture. At 78% WHC, most of the enchytraeids were found on the surface of the soil, which was flooded.

The relationship between daily feeding activity and relative soil moisture content until peak feeding activity could be described by a simple linear regression for both test species (Fig 3). Differently, when considering the whole datasets, no significant correlation was found (Spearman's tests, p > 0.05). For the relative moisture contents up to the optimum, i.e., from 16% to 52% WHC for *E. andrei* and from 40% to 68% WHC for *E. albidus*, a multiple linear regression, including two

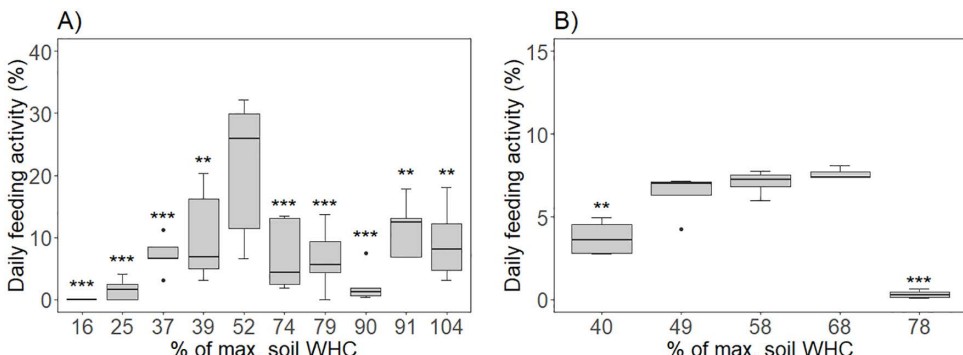

**Fig 2. Daily feeding activity at different relative soil moisture contents for (A) the earthworm, *Eisenia andrei*, and (B) for the enchytraeid, *Enchytraeus albidus*.** Daily feeding activity is expressed as % of consumed bait per day, and relative soil moisture content is expressed as % of maximum water holding capacity (% WHC). For the earthworm tests, n = 10 at 52% WHC and n = 5 at all other relative moisture contents. For the enchytraeid test, n = 3 at 68% WHC, n = 4 at all other relative moisture contents. Significant differences from the reference moisture (i.e., 52% WHC for *E. andrei* and 49% WHC for *E. albidus*) are indicated by * (p < 0.05), ** (p < 0.01) and *** (p < 0.001), according to Dunnet's post-hoc with Holm's correction.

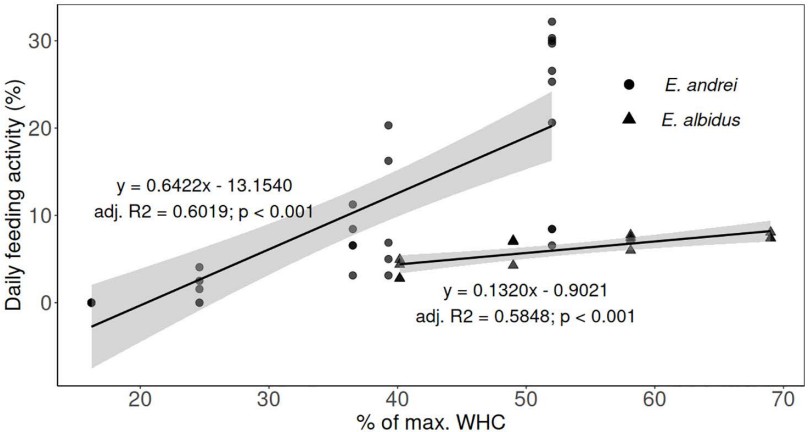

**Fig 3. Linear regressions of the daily feeding activity of each test species exposed to different soil moistures up to their respective peak.** Black lines indicate the linear regression, and the 95% confidence interval is indicated by the grey shades. Daily feeding activity is expressed as % of consumed bait per day, and relative soil moisture content is expressed as % of maximum water holding capacity (% WHC). Each symbol (black dots for *Eisenia andrei* and black triangle for *Enchytraeus albidus*) represents values of daily feeding activity of all replicates (n = 10 at 52% WHC and n = 5 at all other moisture contents for *E. andrei*, n = 3 at 68% WHC and n = 4 for all other moisture contents for *E. albidus*).

fixed effects (species and moisture) and their interaction, showed a significant influence of the interaction term (p > 0.05), indicating that there are differences in moisture effects on feeding activity between the two species (Table 1).

Finally, multiple regression analysis was performed with the regression model obtained in a previous field experiment (Campiche et al. [20]). The linear regression model produced in the mentioned study is shown in Fig 4, in addition to the ones obtained in the present studies. For this comparison, moisture content was expressed as gravimetric water content (mass of water per mass of dry soil, in percentage). The results of the multiple regressions (Table 2) showed no significant differences in the feeding activity response to moisture, when comparing *E. albidus* (this study) with Campiche et al. [20], and a common slope of 0.3384 (0.2824 – 0.3944) was estimated for the two studies. On the other hand, a significant difference was found when comparing *E. andrei* (this study) with Campiche et al. [20] with a significant contribution from the interaction term.

**Table 1. Parameters of the multiple regression analysis for the daily feeding activity response to the two predictors 'moisture' and 'species', with interaction between the two factors.**

| | $y = \beta_0 + \beta_1 \cdot moisture + \beta_2 \cdot species + \beta_3 \cdot moisture \cdot species$ | |
| --- | --- | --- |
| **Coefficient** | **Estimate (95% confidence interval)** | **p value** |
| $\beta_0$ | −13.1540 (−19.4458 − 6.8622) | <0.001 |
| $\beta_1$ | 0.6422 (−0.4811 − 0.8033) | <0.001 |
| $\beta_2$ | 12.2519 (−4.6932 − 29.1969) | 0.1519 |
| $\beta_3$ | −0.5102 (−0.8428 − −0.1776) | < 0.01 |

Data on moisture are expressed as percentage of the maximum water holding capacity (WHC). The two species compared are *Eisenia andrei* and *Enchytraeus albidus*. β0, β1, β2, and β3 are the coefficients of the linear regression. Adjusted $R^2 = 0.6085$.

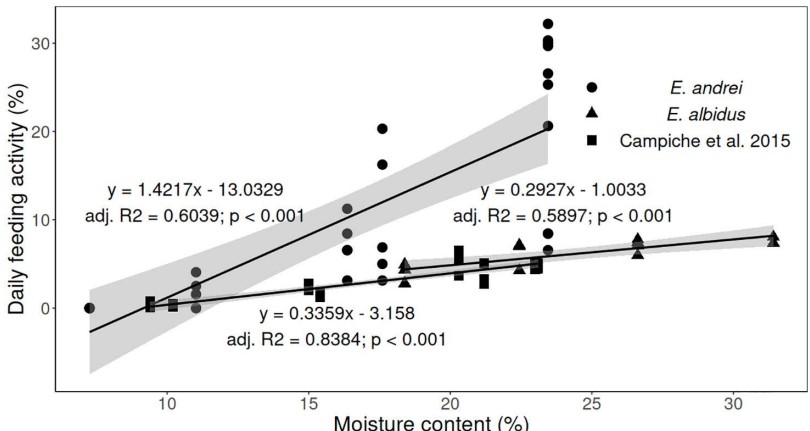

**Fig 4. Linear regressions of the daily feeding activity (in % of pierced holes) of each experiment exposed to different soil moistures (gravimetric water content, %).** Black lines indicate the linear regression, and the 95% confidence interval is indicated by the grey shades. Daily feeding activity is expressed as % of consumed bait per day, and soil moisture content is expressed as % of the soil dry weight). Each symbol (black dots for *E. andrei*, this study, black triangle for *E. albidus*, this study, black squares for Campiche et al. 2015 [20] represents values of daily feeding activity of all replicates (n = 10 at 23% moisture content and n = 5 at all other moisture contents for *E. andrei*, n = 3 at 31% moisture content and n = 4 for all other moisture contents for *E. albidus,* n = 4 for Campiche et al. 2015).

## Discussion

### Test performance and optimal moisture conditions for the two species

Soil in the test replicates was in general drier at the end of the tests compared to the start, with more pronounced differences in the earthworm test. This difference might be due to higher metabolic water requirements of earthworms, which could have contributed to greater soil moisture loss. In addition, in the enchytraeid experiments, soil moisture was monitored and adjusted throughout the longer test duration, whereas this was not done for the shorter earthworm experiments. A humidity check during the test, as suggested in standard guidelines for testing reproduction, is thus recommended for the bait lamina test in the laboratory for both species even in case of short test duration.

The bait lamina test is rarely applied under extreme soil moisture conditions. Its suitability in saturated media has recently been demonstrated through sediment assays [16] highlighting the potential of this method to assess organic matter breakdown in sediments. In the present study, no bait loss was recorded in negative control replicates without

**Table 2. Parameters of the multiple regression analysis for the daily feeding activity response to the two predictors 'moisture' and 'experiments', with 1) interaction between the two factors and with 2) no interaction.**

**1) $y = \beta_0 + \beta_1 \cdot moisture + \beta_2 \cdot experiment + \beta_3 \cdot moisture \cdot experiment$**

| | *E. andrei* VS Campiche et al. [20]. Adjusted $R^2 = 0.6759$ | | *E. albidus* VS Campiche et al. [20]. Adjusted $R^2 = 0.8491$ | |
|---|---|---|---|---|
| Coefficient | Estimate (95% confidence interval) | p value | Estimate (95% confidence interval) | p value |
| $\beta_0$ | −13.0329 (−18.2403 – −7.8254) | < 0.001 | −1.003 (−3.6296 – 1.6231) | 0.445 |
| $\beta_1$ | 1.4217 (1.1252 – 1.7181) | < 0.001 | 0.2927 (0.1865 – 0.3990) | < 0.001 |
| $\beta_2$ | 9.8474 (1.9266 – 17.7682) | < 0.05 | −2.1822 (−5.0632 – 0.6989) | 0.134 |
| $\beta_3$ | −1.0657 (−1.5111 – −0.6203) | < 0.001 | 0.0633 (−0.0618 – 0.184) | 0.313 |

**2) $y = \beta_0 + \beta_1 \cdot moisture + \beta_2 \cdot experiment$**

| | *E. albidus* VS Campiche et al. [20]. Adjusted $R^2 = 0.8486$ | | | |
|---|---|---|---|---|
| Coefficient | Estimate (95% confidence interval) | p value | | |
| $\beta_0$ | −2.1118 (−3.5594 – −0.6643) | < 0.01 | | |
| $\beta_1$ | 0.3384 (0.2824 – 0.3944) | < 0.001 | | |
| $\beta_2$ | −0.7709 (−1.4906 – −0.0511) | < 0.05 | | |

Data on moisture are expressed as gravimetric moisture content (%). The experiments compared are *Eisenia andrei* (this study), *Enchytraeus albidus* (this study), and Campiche et al. [20]. β0, β1, β2, and β3 are the coefficients of the linear regressions.

test organisms with the only exceptions of the two highest tested moistures (nominal 80% and 100% WHC), where a slight bait loss was observed (see SI2). This loss was more likely due to bait dissolution in water but remained minimal (≤ 2%), suggesting that the bait lamina test is applicable across a wide range of soil moisture contents, from very dry to oversaturated.

Both *E. andrei* and *E. albidus* were confirmed to be efficient feeders of the bait material, with average daily feeding activity of 22% and 6% at the respective reference moisture contents. As biomass per replicate was comparable across experiments, the results indicate that *E. andrei* was about three times faster than *E. albidus*.

For earthworms, feeding activity was highest (i.e., 22% per day) at the reference moisture content (i.e., 52% WHC), suggesting that the optimal conditions for the bait lamina test, fall within the range recommended for *E. fetida/andrei* reproduction assays (40 – 60% WHC) [22,23] and are close to the value recommended for testing avoidance behavior (60% WHC) [21]. Daily feeding activity at the reference soil moisture was comparable to values reported in other laboratory experiments with *Eisenia andrei/fetida* at similar soil moisture contents. Van Gestel et al. [7] exposed ten earthworms in approximately 814 g of OECD artificial soil with 5 bait lamina strips and estimated that the time required to empty 50% of the holes corresponded to approximately 4.6 days. Casabé et al. [14] obtained an average feeding activity of 62.8% after three days (i.e., 22.7% of consumed bait per day) by exposing six earthworms in 350–400 g of uncontaminated clay silty soil with 4 bait lamina sticks. Finally, Jänsch et al. [15] obtained more than 30% feeding activity after seven days when exposing 10 earthworms in artificial and eight natural soils with 2 bait strips. When normalized to test duration, number of earthworms, and number of bait sticks used, the results of feeding activity correspond to 0.95 [14]; 0.46 (this study); 0.22 [7]; and 0.21 [15] % of consumed bait per day per earthworm per bait stick. Mesocosms or terrestrial model ecosystems with natural soil [4–6] suggested as well a high contribution of earthworms to bait consumption, but overall feeding activity was in general lower compared to values observed in the above mentioned laboratory experiments. This difference can be explained by both the presence of other species than *E. andrei/fetida* and the greater complexity of test conditions, which more closely reflect field situations. In particular, the use of intact soil cores preserves soil structure, whereas in classical single-species laboratory experiments, soil sieving leads to structure loss. Additional factors can be the different intrinsic properties of the tested soil, such as organic matter content, texture, or presence of other food

sources that could have been preferred over the bait material. Finally, interactions with other soil organisms, such as other mesofauna or microbial communities, may further influence earthworm feeding behavior, for example through resource competition or microbial-mediated changes in nutrient availability.

For enchytraeids, optimal feeding activity occurred between the reference moisture (49% WHC) and 68% WHC (6–7% per day), suggesting that the range recommended in the standard guidelines (between 40 and 60% of maximum WHC [24,25]) is also suitable for bait lamina tests. The wider optimal range for feeding activity for enchytraeids compared to earthworms might suggest a higher tolerance for greater levels of moisture contents. Like for earthworms, our experiments confirmed that enchytraeids are important consumers of the bait, which is aligned with findings of other laboratory studies. Helling et al. [28] exposed 75 adults enchytraeids (*E. minutus* and *E. lacteus*) in 75 g of arable soil and artificial OECD soil for 10 days with one bait lamina stick and obtained 17 and 7% of feeding activity, respectively. Bart et al. [13] observed approximately 70% of feeding activity after 5 days of exposure of 12 adult *E. albidus* in 50 g of a luvisol with 1 stick. When normalized to test duration, number of enchytraeids, and number of bait sticks used, the results of feeding activity correspond to 1.16 [13]; 0.23 (arable soil) and 0.01 (OECD soil) [28]; and 0.01 (this study) % of consumed bait per day per enchytraeid per bait stick. Even after accounting for this normalization, the studies are difficult to compare because different densities of organisms per unit of soil were used. Interestingly, density alone doesn't seem to fully explain the observed differences: the study with the highest density [28] reported feeding rate more comparable to our results whereas in the study with the lowest density a higher bait consumption was observed [13]. One possible reason for this is that in the latter case, the test organisms were distributed across different soil depths, which could have facilitated access to the bait stick. Differently, a laboratory experiment, showed very low feeding activity of the enchytraeid *Cognettia sphagnetorum* which was exposed in a forest soil, under a wide range of soil moistures and temperatures [4]. Such lower activity compared to other laboratory experiments might be explained by the very high content in soil organic matter, which could have been a preferred source of food for those enchytraeids rather than the bait. A similar hypothesis was formulated by André et al. [29] who reported a significant negative correlation between soil organic matter content and feeding activity for a site characterized by a thick litter layer.

## Moisture effect on feeding activity

For both test species, feeding activity was low under dry conditions and increased with increasing soil water contents until reaching a peak feeding activity at the species-specific optimum moisture levels. These results are in line with the general knowledge that moist soils are more favorable for earthworm activity [30] and that, similarly, enchytraeids are sensitive to drought [31–33].

Differently, the effect of very high moisture contents differed between the two species. For earthworms, feeding activity declined beyond the optimum (52% WHC), but remained relatively stable with no clear correlation with soil water content. Although excessive moisture can impair earthworm performance [34], many species are able to survive even for long periods in completely submerged soils [30]. Similar observations could be found in our study, where *E. andrei* survived at all relative moisture contents up to 104% WHC, albeit with reduced feeding activity compared to the optimum. For enchytraeids, only one moisture treatment was higher than the optimal range (49 – 69% WHC) and thus there are not enough data available to investigate possible correlations between feeding activity and high soil moisture. At this highest moisture treatment, enchytraeids feeding activity was almost absent. In this treatment, the soil appeared flooded and many enchytraeids, although alive and active, remained at the surface, suggesting reduced burrowing ability and limited access to the bait strips. Thus, *E. albidus* appeared more strongly affected than *E. andrei* by excessive moisture conditions.

The increase in feeding activity with increasing soil moisture could be modelled by linear regressions but only up to the peak. The modeled range is expected to reflect the conditions most suitable for monitoring feeding activity, generally considered to lie between slightly moist and field capacity [35]. For loamy soils such as LUFA 2.2, plant wilting occurs below

7 % gravimetric water content, while field capacity is around 20% [36]. The models derived for earthworms and enchytraeids cover gravimetric content ranges of 7 – 23% and 18 – 31% gravimetric water content, respectively (see SI2). It can therefore be suggested that both models would be relevant for optimal monitoring conditions of soil moisture contents, for sandy loamy soils at expected field capacity. However, care should be taken when comparing field soils and sieved soils because of possible changes due to structure loss and further testing is needed to verify this hypothesis. Additionally, the current models would need to be validated for soil types other than sandy loam.

The multiple regression analysis showed a significant interaction between species and soil moisture content, indicating that the increase in feeding activity over the defined soil moisture range was significantly different between the two species. This increase was more pronounced for *E. andrei* compared to *E. albidus*, which is not surprising, as the first has been shown to feed on the bait faster than the latter. Different factors could explain the faster bait consumption of *E. andrei* compared to *E. albidus*. A biological explanation is that earthworms inherently exhibit faster feeding rate than enchytraeids. A mechanical explanation can be that earthworms, being bigger than enchytraeids, have more facility to break and feed on the bait material. Indeed, enchytraeids and earthworms play similar functional roles, but at different spatial scales [37]. The difference in size may also allow earthworms to move around in the soil more easily than enchytraeids, enabling them to access the sticks more quickly. Finally, the intrinsic habitat quality of the LUFA 2.2 may have contributed. If alternative food sources were limited for *E. andrei*, they may have relied more heavily on the bait material, whereas *E. albidus* may have exploited more soil inherent resources and thus consuming the bait more slowly. This last hypothesis could be supported with the fact that earthworms and enchytraeids have been suggested to have slightly different ecological preferences and niches [38,39].

The model developed for *E. albidus* in this study showed a more similar trend, compared to *E. andrei*, to the model developed in the field study Campiche et al. [20]. It is worth mentioning that the species used in this study (*E. andrei*) is not a relevant representative of agricultural fields [37]. Moreover, earthworms can have different feeding behaviors depending on which ecological group they belong. Epigeic species, such as *E. andrei*, live in and feed on the litter layer, where they initiate the decomposition process of complex organic material. On the other hand, endogeic species prefer pre-processed organic material, while anecic earthworms mainly transport organic litter from the soil surface to deeper soil layers [40]. The extent to which ecological groups contribute differently to bait consumption remains unknown. From our results, it can be suggested that epigeic earthworm species are not the best representative of the agricultural field studied by Campiche et al. [20], whereas *E. albidus* seems to better reflect the biological activity of the field. This is in line with some observations suggesting that in agricultural ecosystems *Enchytraeus* species are generally more abundant [41] and more ecologically relevant than earthworms, such as *E. andrei* [37,42]. It cannot be excluded that other species and/or ecological categories than *E. andrei* and/or epigeic species were present and active in the field from Campiche et al. [20], and that they may have had a smaller contribution to bait consumption. For each site studied, further investigations would be needed to assess the real soil community and draw more consistent conclusions on the relative contribution of each specific soil organism group to feeding activity, measured through the bait lamina test.

Globally, the results of our study are an important first step for normalizing bait-lamina test data between studies and sites based on soil moisture, which is one of the biggest confounding factors in field testing. To further validate the produced model, additional factors should still be explored, which can have an influence on feeding activity under field conditions. Intrinsic soil properties, such as texture or organic matter content, can strongly influence water retention and therefore the activity of soil organisms [28,43]. The bait lamina test should be performed with other soil types than the LUFA 2.2 to explore potential differences in the model. In addition, the soil used in laboratory experiments is generally sieved, disrupting the original structure that can be found under natural field conditions. The loss of structure can have an important influence on the capacity of the soil to retain water and therefore typical characteristic such as wilting point and field capacity might be less comparable. Environmental factors (e.g., temperature) are also known to play a role on the behavior of the soil community as well as soil conditions. Finally, natural communities of soil organisms are more diverse

and complex than the limited number of species that can be tested in the laboratory and this can influence the velocity of bait consumption as well as the main feeders of the bait, as already highlighted by the two tested species.

## Conclusions

The bait-lamina method has been shown to be a suitable tool for evaluating the impact of soil moisture on feeding activity of two soil species used in standard testing. For both earthworms and enchytraeids, feeding activity increased linearly with increasing soil moisture up to a peak. In addition to species differences in feeding rates and optimum moistures, the linear increases in feeding activity were also found to be species dependent. The linear model was able to describe the feeding response of two important soil invertebrate species for a realistic range of soil moisture contents. The produced models will facilitate interpretation of future field studies, but first additional studies are required to validate this model under complex field conditions.

## Supporting information

**S1 File. Supporting Information.**
(DOCX)

## Acknowledgments

We would like to thank INRAE/AgroParisTech (UMR ECOSYS, Versailles, France) for having kindly provided the species *Enchytraeus albidus* Henle 1847. We would also like to thank the Statistical Consulting service, at the ETH of Zurich, for their help with the statistical analysis.

## Author contributions

**Conceptualization:** Sophie Campiche, Janine W. Y. Wong, Benoit J. D. Ferrari.

**Formal analysis:** Gilda Dell'Ambrogio, Mathieu Renaud.

**Methodology:** Gilda Dell'Ambrogio, Sophie Campiche.

**Project administration:** Sophie Campiche.

**Supervision:** Inge Werner, Benoit J. D. Ferrari.

**Writing – original draft:** Gilda Dell'Ambrogio, Sophie Campiche, Janine W. Y. Wong, Christina Lüthi, Inge Werner, Benoit J. D. Ferrari.

**Writing – review & editing:** Gilda Dell'Ambrogio, Sophie Campiche, Janine W. Y. Wong, Mathieu Renaud, Christina Lüthi, Inge Werner, Benoit J. D. Ferrari.

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
