## [Decision Letter · Decision Letter 0]

1 Aug 2025

Dear Dr. Dell'Ambrogio,

We look forward to receiving your revised manuscript.

Kind regards,

Michael Schubert

Academic Editor

PLOS ONE

Journal Requirements:

2. Please amend your list of authors on the manuscript to ensure that each author is linked to an affiliation. Authors’ affiliations should reflect the institution where the work was done (if authors moved subsequently, you can also list the new affiliation stating “current affiliation:….” as necessary).

Reviewers' comments:

Reviewer's Responses to Questions

**Comments to the Author**

1. Is the manuscript technically sound, and do the data support the conclusions?

Reviewer #1: No

Reviewer #2: Yes

2. Has the statistical analysis been performed appropriately and rigorously?

Reviewer #1: No

Reviewer #2: Yes

3. Have the authors made all data underlying the findings in their manuscript fully available?

Reviewer #1: Yes

Reviewer #2: Yes

4. Is the manuscript presented in an intelligible fashion and written in standard English?

Reviewer #1: No

Reviewer #2: Yes

Reviewer #1: The paper describes a laboratory experiment involving bait-lamina and the effect of soil moisture on the feeding activities of two soil animal species. While the subject matter is potentially interesting for readers, the manuscript cannot be published due to the weak presentation of the data and errors in the statistical analysis.

The most important analysis is missing - to what extent are 52% and 68% for the two species statistically significantly different?

The text is too long-winded. The style needs to be changed to be more concise, with excessive words and repetitions removed.

It is quite surprising, that earthworms contribute in feeding activity on bait lamina strips – they are rather bigger soil animals. Maybe Authors used very small (young) individuals? Please specify animals age/size. Are you sure that the earthworm treatments have not been “contaminated” with enchytraeids?

L163 why two tests? Please combine it and just mark, that experiment was performed in two stages

L171 why were the same values not used as for earthworms? Such results do not compare well

L194 was the biomass of earthworms and enchytraeids comparable? There is difficult to find these information in the text. Moreover, moisture effect may depend on animals density.

L224 why was it done? Rinsighted strips can wash out the medium, making such a measurement pointless

Tab.1 I do not see the point of stating these values. Moreover, table is too big and unclear

L259 combining two species in this analysis has no sense, they do not interact with each other

Reviewer #2: The article submitted by Dell’Ambrogio et al. addresses a technique—the use of Bait Lamina—that provides a simple and low-cost method for assessing soil functional quality. As such, it represents a promising tool for evaluating the functional impacts of various environmental stressors on this key ecosystem compartment. However, its use remains relatively limited, leading to a knowledge gap that sometimes hampers the interpretation of results. In this context, the study by Dell’Ambrogio et al. appears both relevant and important for improving the robustness and applicability of future research employing this technique.

The study, conducted exclusively under laboratory conditions using a single soil type and two model organisms, appears to have been carried out in a rigorous manner, following standardized protocols (number of tested conditions, number of replicates, number of individuals, etc.) that allow for robust statistical analysis and ensure the scientific reliability of the results. The findings are original and respond to a clearly identified knowledge need.

Therefore, this article may be considered suitable for publication in PLOS ONE.

However, I question the relevance of so strongly associating this study with the aim of improving in situ use of bait lamina. The experimental conditions implemented in this study are quite distant from those encountered in field situations, both in terms of the organisms present and the physico-chemical properties of the soils (which are not sieved in situ and may span a wide range of pedological conditions). The authors are aware of this and have discussed it in the manuscript. Nonetheless, I believe that the stated objective of the study should be more aligned with the actual scope of the results obtained. In other words, there is no need to overstate the findings, which are valuable in their own right even without a direct connection to field conditions.

For example, I am personally not convinced by the comparison made between the present study’s results and those from previous in situ work by Campiche et al. (Fig. 4). In my view, linking these two sets of results is too speculative—despite the tempting similarity in trends observed with E. albidus—given the many differences between the two studies (which, again, are well discussed by the authors). In my opinion, this comparison should be used solely to feed the discussion, especially to highlight perspectives and the need for further research to better interpret in situ responses. However, if the authors decide to maintain this comparison, it seems essential to provide more information about the data from Campiche et al., to better assess to what extent those results are generalizable across diverse environmental conditions.

Beyond this general comment, which reflects my main concern about the current manuscript, here are a few additional—mostly minor—comments that may help improve the clarity and quality of the manuscript:

Abstract

• In light of the above comment, I recommend revising the abstract to reduce the emphasis on in situ applications.

Highlights

• Similarly, the final bullet point seems too speculative to be featured as a highlight.

• Also, the phrase “field situation” is too vague and should be clarified or removed.

• Some highlights appear too long—please verify that they meet the journal’s character limits.

Introduction

• L.70–71: The word “soil” appears too frequently in the same sentence—please try to avoid this repetition.

• L.94: Replace “at the Ecotox Centre” with “Campiche et al.”

• L.95–97: The information provided here about the work by Campiche et al. is insufficient to allow the reader to fully assess its scope or the generalizability of its findings.

Materials & Methods

• L.161–162: Avoid future tense (“will be expressed”).

• L.163–175: Why is the number of replicates different between the two biological models? This would be helpful to clarify for readers.

• L.220: The phrase “once a week” is odd given that the test lasted only 12 days. Consider rephrasing as “at most once.”

Results

• L.265: How can the high mortality rate observed in earthworms at 25% WHC be explained? Unless I missed it, this does not appear to be discussed.

• Table 1: What does the number 4 in the final column of the last row refer to?

• Figures 3 & 4: Rather than showing only the means, would it not be more informative to display all replicates to better convey data variability?

• Figure 4: How many replicates were considered to calculate the means from Campiche et al.? Were these replicates all from the same soil or different soils? Such information is largely missing in the current integration of this prior work.

Discussion

• L.370–371: The explanation that lower variation in moisture in the enchytraeid test is due to moisture adjustments is not very convincing, especially considering the difference in test duration (2 vs. 12 days).

• L.416: What exactly is meant by “interaction within soil communities,” and how could this influence bait lamina feeding? This point should be expanded.

• L.433: The question of soil quantity should be discussed in terms of density per unit of soil. Were densities higher in the studies cited compared to the present one? Has this hypothesis been tested in previous research?

• L.474: While the point about “sandy loamy soils” is valid, what about other soil types? This should be addressed to better define the study’s limitations.

• L.484–485: Could mobility also be a biological factor explaining the differences in bait lamina feeding efficiency?

• L.498–500: Due to the significant differences between lab and field conditions (unsieved soil, natural species different from lab models, etc.), this hypothesis appears too speculative in its current form. The same applies to lines 521–522 and 541–544.

.

Reviewer #1: No

Reviewer #2: No

---

## [Author Response · Author response to Decision Letter 1]

4 Oct 2025

The main criticisms concerned the statistical analysis, the language, and the conclusions.

We adapted the statistical analysis by referring to a range of optimal moisture contents for enchytraeids rather than a single peak, while keeping the rest of the analysis unchanged, as we believe it remains sound. We revised the language to improve conciseness, clarity, and flow, and we removed overly speculative hypotheses that oversimplified the link between laboratory and field results.

Please see the attached Response to Reviewers document for a detailed, point-by-point reply to all comments. We believe these modifications have strengthened the manuscript and made it suitable for publication.

---

## [Decision Letter · Decision Letter 1]

26 Jan 2026

Dear Dr. Dell'Ambrogio,

Thank you for submitting your manuscript to PLOS ONE. After careful consideration, we feel that it has merit but does not fully meet PLOS ONE’s publication criteria as it currently stands. Therefore, we invite you to submit a revised version of the manuscript that addresses the points raised during the review process. Note that the reviewer voiced serious concerns about the quality of the revisions implemented in the revised manuscript. Should these concerns persist in the next round of peer review, the manuscript will not be acceptable for publication in PLOS ONE.

We look forward to receiving your revised manuscript.

Kind regards,

Michael Schubert

Academic Editor

PLOS One

Journal Requirements:

Reviewer's Responses to Questions

**Comments to the Author**

Reviewer #1: (No Response)

2. Is the manuscript technically sound, and do the data support the conclusions?

Reviewer #1: No

3. Has the statistical analysis been performed appropriately and rigorously?

Reviewer #1: No

4. Have the authors made all data underlying the findings in their manuscript fully available?

Reviewer #1: Yes

5. Is the manuscript presented in an intelligible fashion and written in standard English?

Reviewer #1: Yes

Reviewer #1: Although the article deals with an interesting and important issue, it is very poorly written. In fact, every sentence raises doubts, and the revisions made little difference to the overall text. I admit that I didn't read it to the end because the changes are merely cosmetic. The authors must put in much more work to make the text understandable and interesting to readers. Of course it's the Editor's decision, but the text in the current version falls far short of the standard of other papers in PlosOne.

L1 please change ‘soil oligochaetes’ into two tested species names

L20-21 remove this phrase ‘commonly performed under field conditions to measure the impact of chemicals on soils’

L21-24 combine these two sentences in one, more concise

L27 add ‘standard soil’ and add producer name

L44 model validation is not a scientific goal

L41-46 Do the authors mean watering the strips installed in the field?

L73 reference 5 refers to something completely different

L74-80 This is not important in the context of the topic of the work—I suggest focusing more on the ecological applications of the method.

L83 not only in field conditions, these applied also to laboratory applications

L119-121 Why was this done? It could have affected the microbiological decomposition of the medium.

L147-151 Did the authors measure these parameters, or were they provided by the manufacturer?

L149 double parenthesis

L156-8 Why were the soils treated differently? This could have affected the results, albeit indirectly, and the impact of animals and freezing was measured in an interaction that was difficult to determine.

L166-167 This should be relocated to earlier section

L169-176 What is the point of these calculations if the authors used only one soil type?

L179 60% is not intermediate comparing to first series

It is completely unclear what the comparison with the Campiche et al. model is supposed to mean.

.

Reviewer #1: No

---

## [Author Response · Author response to Decision Letter 2]

3 Mar 2026

We have reviewed the detailed comments where possible. However, as some general points remain open, we have submitted a separate author query requesting guidance on how to address these general comments. We look forward to your advice.

---

## [Editor Report · Decision Letter 2]

5 Mar 2026

Feeding activity of Eisenia andrei and Enchytraeus albidus under different soil moisture regimes assessed by the bait-lamina test

PONE-D-25-35367R2

Dear Dr. Dell'Ambrogio,

We’re pleased to inform you that your manuscript has been judged scientifically suitable for publication and will be formally accepted for publication once it meets all outstanding technical requirements.

Kind regards,

Michael Schubert

Academic Editor

PLOS One

---

## [Editor Report · Acceptance letter]

PONE-D-25-35367R2

PLOS One

Dear Dr. Dell’Ambrogio,

I'm pleased to inform you that your manuscript has been deemed suitable for publication in PLOS One. Congratulations! Your manuscript is now being handed over to our production team.

Kind regards,

on behalf of

Dr. Michael Schubert

Academic Editor

PLOS One